# Aquaporin-7-Mediated Glycerol Permeability Is Linked to Human Sperm Motility in Asthenozoospermia and during Sperm Capacitation

**DOI:** 10.3390/cells12152003

**Published:** 2023-08-05

**Authors:** João C. Ribeiro, Raquel L. Bernardino, Ana Gonçalves, Alberto Barros, Giuseppe Calamita, Marco G. Alves, Pedro F. Oliveira

**Affiliations:** 1Unit for Multidisciplinary Research in Biomedicine (UMIB), Institute of Biomedical Sciences Abel Salazar (ICBAS), University of Porto, 4050-313 Porto, Portugal; up202009612@up.pt (J.C.R.); raquellbernardino@gmail.com (R.L.B.); 2Laboratory for Integrative and Translational Research in Population Health (ITR), University of Porto, 4050-313 Porto, Portugal; 3LAQV-REQUIMTE and Department of Chemistry, University of Aveiro, 3010-193 Aveiro, Portugal; 4Centre for Reproductive Genetics Professor Alberto Barros, 4100-012 Porto, Portugalabarros@med.up.pt (A.B.); 5Department of Pathology, Faculty of Medicine, University of Porto, 4200-319 Porto, Portugal; 6i3S-Instituto de Investigação e Inovação em Saúde, University of Porto, 4200-135 Porto, Portugal; 7Department of Biosciences, Biotechnologies and Environment, University of Bari “Aldo Moro”, 70125 Bari, Italy; giuseppe.calamita@uniba.it; 8iBiMED-Institute of Biomedicine, Department of Medical Sciences, University of Aveiro, 3810-193 Aveiro, Portugal; alvesmarc@gmail.com

**Keywords:** aquaglyceroporins, male fertility, sperm capacitation, sperm motility, spermatozoa

## Abstract

Osmoregulation plays a vital role in sperm function, encompassing spermatogenesis, maturation, and fertilization. Aquaglyceroporins, a subclass of aquaporins (AQPs), facilitate the transport of water and glycerol across the sperm membrane, with glycerol serving as an important substrate for sperm bioenergetics. This study aimed to elucidate the significance of AQP-mediated glycerol permeability in sperm motility. The presence and localization of AQP3 and AQP7 in human sperm were assessed using immunofluorescence. Subsequently, the glycerol permeability of spermatozoa obtained from normozoospermic individuals (*n* = 30) was measured, using stopped-flow light scattering, after incubation with specific aquaporin inhibitors targeting AQP3 (DFP00173), AQP7 (Z433927330), or general aquaglyceroporin (phloretin). Sperm from asthenozoospermic men (*n* = 30) were utilized to evaluate the AQP7-mediated glycerol permeability, and to compare it with that of normozoospermic men. Furthermore, hypermotile capacitated sperm cells were examined, to determine the AQP7 expression and membrane glycerol permeability. AQP3 was predominantly observed in the tail region, while AQP7 was present in the head, midpiece, and tail of human sperm. Our findings indicate that AQP7 plays a key role in glycerol permeability, as the inhibition of AQP7 resulted in a 55% decrease in glycerol diffusion across the sperm membrane. Importantly, this glycerol permeability impairment was evident in spermatozoa from asthenozoospermic individuals, suggesting the dysregulation of AQP7-mediated glycerol transport, despite similar AQP7 levels. Conversely, the AQP7 expression increased in capacitated sperm, compared to non-capacitated sperm. Hence, AQP7-mediated permeability may serve as a valuable indicator of sperm motility, and be crucial in sperm function.

## 1. Introduction

Fertility rates have been decreasing in the last decades in developed countries [1]. According to the WHO guidelines, one of the parameters used to assess male fertility is the motility percentage of the ejaculated sperm [2]. Sperm cells acquire their motility during the maturation process in the epididymis that allows the sperm cells to move through the female reproductive tract, and to develop the state of hypermotility in the final stages of their journey, an activation called capacitation. During the journey of the sperm cellls through the male and female reproductive tract, mediumosmolarity is in constant change, thus osmoregulation is important for proper sperm maturation and fertilization [3,4,5,6].

Aquaporins (AQPs) are transmembrane proteins that assist in the diffusion of water and small neutral solutes through the plasma membrane. Humans possess 13 different AQPs, which are generally divided into three different subclasses: orthodox (AQP0, AQP1, AQP2, AQP4, AQP5, AQP6, AQP8), aquaglyceroporins (AQP3, AQP7, AQP9, AQP10), and superaquaporins (AQP11, AQP12). Human sperm cells are known to express AQP3, AQP7, AQP8, and AQP11 [7,8,9,10,11]. It is worth mentioning that two of the four different AQPs expressed are aquaglyceroporins (AQP3 and AQP7), which are also capable of allowing the diffusion of small solutes, such as glycerol [12]. The role of glycerol in male fertility is still a mystery. However, it has been hypothesized that this polyol could have an important role in epididymal sperm maturation, and be utilized by sperm cells as an energy source [13,14,15,16]. In fact, aquaglyceroporin expression in the epididymal epithelial cells has been suggested as a marker of male fertility (for review, see: [12]). If this is the case, then the significance of glycerol and its permeability in sperm function and, consequently, male fertility, demands awareness and care, considering that sperm motility is acquired during the epididymal transit. However, the understanding of this relationship remains incomplete, and requires further investigation. Thus, this work aims to unravel the relationship between AQP-mediated glycerol permeability and sperm motility, with a particular focus on the acquisition of hypermotility during sperm capacitation.

## 2. Materials and Methods

### 2.1. Chemicals

Specific inhibitors for AQP3 and AQP7 (DFP00173 and Z433927330, respectively) were purchased through MedChemExpress (Monmouth Junction, NJ, USA). All other chemicals were purchased from Merck KGaA (Darmstadt, Germany), unless stated otherwise.

### 2.2. Patient Characterization and Study Design

Human sperm samples were collected from Centro de Genética da Reprodução Professor Alberto Barros, located in Porto, Portugal, after approval by the Joint Ethics Committee CHUP/ICBAS (2021/CE/P002[P342/CETI/ICBAS]), between September 2020 and July 2022. All patients included in this study signed informed written consent forms. Semen samples were obtained from male patients through masturbation after 2–4 days of abstinence, and were placed in sterile tubes. Additionally, patients were asked to provide information on their consumption of tobacco, alcohol, and other substances that could interfere with the experiment. The fresh semen samples were left to liquefy at 37 °C for at least 30 min, and were then used for sperm parameter measurements. Afterward, sperm cells were centrifuged at 500× *g* for 5 min at room temperature. The seminal fluid was discarded, and the pellet was washed with phosphate-buffered saline (PBS) solution, with the osmolarity corrected to 300 mOsm. After two series of washing cycles performed at 500× *g* for 5 min, the supernatant was discarded, and the pellet was finally resuspended in PBS. The sperm samples were characterized according to the WHO guidelines for the laboratory examination and processing of human sperm [2]. Samples with the following semen parameters, according to the WHO criteria, were utilized in this study: sperm concentration > 15 million/mL, and sperm viability >56%. After this selection, samples were further classified as normozoospermic when the total motility was ≥40%. These samples were used to study the role of each aquaglyceroporin on the sperm glycerol permeability. To do so, the sperm cells were resuspended in an isotonic medium, corrected to 300 mOsm, comprising 137 mM NaCl, 2.7 mM KCl, 10 mM Na_2_HPO_4_, and 1.8 mM KH_2_PO_4_, and 1 mM PMSF at a concentration of 20–30 million sperm cells per mL, and left for 10 min for stabilization. Then, sperm cells from 30 normozoospermic men were incubated for 10 min, with a specific AQP3 inhibitor (iAQP3, 200 nM of DFP00173), a specific AQP7 inhibitor (iAQP7, 200 nM of Z433927330), a general inhibitor of the mediated transport of glycerol phloretin (PHL, 0.35 mM), or the vehicle (DMSO at 1%) [17,18]. Sperm cells from 30 asthenozoospermic men (when the total motility was <40%) were also collected (Table 1), and incubated in the presence and absence of iAQP7, for comparison with the samples from normozoospermic men. Then, the sperm membrane glycerol permeability was measured using a stopped-flow light scattering apparatus, as described below, and the AQP7 protein content was determined using Western blotting, as described below.

### 2.3. Sperm Motility and Viability Measurement

Human sperm motility was measured with the assistance of a Makler counting chamber, pre-heated at 37 °C. An aliquot of 10 µL of sperm suspension was removed from the semen, to access the motility percentage, and proceed with the categorization of normo- or asthenozoospermic samples (Table 1). Likewise, 10 µL of sperm cell solution was used to measure the motility percentage after treatment with the different inhibitors. A total of 100 sperm cells were counted per sample/condition, and the number of sperm with tail movement was regarded as motile sperm, thus including progressive and non-progressive motility. The sperm cells were filmed using a Nikon Eclipse C*i* microscope (Nikon, Shinagawa, Tokyo, Japan). Video was obtained using NIS-Elements Imaging Software (Nikon, Shinagawa, Tokyo, Japan).

The percentage of viable and nonviable sperm cells was determined according to an established protocol [19]. Briefly, 10 µL of semen was mixed with an equal volume of eosin–nigrosin solution, and then was used to conduct smears. A total of 100 sperm cells were counted per slide, in continuous and random fields, under a bright-field optical microscope. Pink-stained sperm cells were considered nonviable, whereas white sperm cells were considered viable.

### 2.4. Spermatozoa Gradient Separation

To unravel the possible role of glycerol permeability in sperm motility, we separated the highly motile sperm from the sperm that were immature or showed a low motility. A total of 12 semen samples from normozoospermic men were used for separation with the GemsR Sperm Wash Gradient Set (45% and 90%) (SWG-01, Genea, Sydney, Australia), used according to the manufacturer’s instructions. In brief, 1.5 mL of 90% solution was dropped into a 15 mL tube, and flowed with an equal volume of 45% solution and semen. Afterward, sperm cells were separated via a centrifugation of 300× *g* for 20 min. High-motile sperm were extracted from the 95% fraction, and low-motile sperm were extracted from the 45% fraction. Each fraction was washed in PBS two times, for further usage in the glycerol permeability measurement through stopped-flow light scattering.

### 2.5. Spermatozoa Capacitation

To capacitate the sperm cells, spermatozoa from normozoospermic samples (N = 12) were washed and then incubated in the presence or absence of a capacitation medium. The capacitation medium comprised NaCl 94.5 mM, KCl 4.8 mM, CaCl_2_ 1.7 mM, KH_2_PO_4_ 1.17 mM, MgSO_4_ 1.22 mM, HEPES 20 mM, glucose 11 mM, NaHCO_3_ 25 mM, and BSA 0.4 mM. Non-capacitation spermatozoa were obtained by placing cells in a similar medium, without NaHCO_3_ and BSA. The spermatozoa were incubated for 3 h at 37 °C, and the capacitation was confirmed by evaluating the total tyrosine phosphorylation (Tyr-P) levels through Western blot analysis. Capacitated and non-capacitated sperm were used in assessing the AQP7 quantification via Western blotting, and the glycerol permeability via stopped-flow light scattering.

### 2.6. Immunofluorescence

Immunofluorescence was performed to further observe the cellular localization of AQP3 and AQP7 in human sperm. Sperm cells from a normozoospermic healthy donor were washed and smeared on a slide, and left to dry. Then, the sperm cells were fixed with 10% paraformaldehyde for 30 min, and permeabilized via incubation with a solution of 0.1% Triton X-100 in PBS, both at room temperature. The sperm cells were blocked in 0.1% gelatine in PBS for 60 min at 37 °C, and incubated with the primary antibodies anti-AQP3 (1:50, ab125219, Abcam, Cambridge, UK), and mouse anti-AQP7 (1:50, sc-376407, Santa Cruz Biotechnology, Heidelberg, Germany) overnight at 4 °C. Cells were incubated without primary antibodies as the negative control. Afterward, the cells were washed and incubated with Alexa Fluor 488 goat anti-mouse IgG (1:1500, A-11001, Thermo Fischer Scientific, Waltham, MA, USA), or Alexa Fluor 488 goat anti-rabbit IgG (1:1500, A-11008, Thermo Fischer Scientific, Waltham, MA, USA) for 1 h at room temperature. Coverslips were mounted, and the nuclei were stained with VECTASHIELD^®^ Antifade Mounting Medium with DAPI (Vector Laboratories, Burlingame, CA, USA). Images were obtained using Zen Black Software (Carl Zeiss, Jena, Germany). For the visualization of the slides, the fluorescence was observed using a Nikon Eclipse C*i* microscope (Nikon, Shinagawa, Tokyo, Japan) equipped with a CoolLed pE-300 lite (CoolLed, Andover, England). Images were obtained using NIS-Elements Imaging Software (Nikon, Shinagawa, Tokyo, Japan). The specificity of the antibodies used was tested via Western blot (Appendix A). 

### 2.7. Western Blotting

Sperm samples from 20 normozoospermic and 20 asthenozoospermic men, as well as 12 samples of non-capacitated and capacitated sperm from normozoospermic men, were precipitated via centrifugation at 5000× *g* for 5 min. The sperm pellet was resuspended, and its protein was extracted via incubation with a solution of sodium dodecylsulfate at 1% for 30 min, at room temperature, and under constant agitation. After that, the remaining cellular residues were centrifugated via centrifugation at 15,000× *g* for 20 min. The supernatant was collected, and its protein content was quantified using Pierce™ BCA Protein Assay Kit (Thermo Fischer Scientific, Waltham, MA, USA). Western blotting was performed using standard methods [19]. The protein samples (10 µg) plus Laemmli buffer were incubated for 10 min at 37 °C. Then, the protein samples were fractionated on a 10% polyacrylamide gel from the TGX Stain-Free™ FastCast™ Acrylamide Kit (Bio-Rad, Hemel Hempstead, UK), and transferred to a low-fluorescence polyvinylidene difluoride (LF-PVDF) membrane (Bio-Rad, Hemel Hempstead, UK), using a Trans-Blot^®^ Turbo™ System (Bio-Rad, Hemel Hempstead, UK). The membranes were blocked with a solution of 5% bovine serum albumin (BSA) in Tris-buffered saline for 90 min. Then, the membranes were incubated overnight at 4 °C separately, with the primary antibodies rabbit anti-AQP3 (1:1000, ab125219, Abcam, Cambridge, UK), mouse anti-AQP7 (1:500, sc-376407, Santa Cruz Biotechnology, Heidelberg, Germany), and mouse anti-phosphotyrosine (1:1000, 05–32, Merck Millipore, Burlington, MA, EUA). Proteins were separately detected upon incubation using goat anti-mouse (1:5000, AP308P, Merck Millipore, Burlington, MA, EUA) or goat anti-rabbit (1:5000, AP307P, Merck Millipore, Burlington, MA, USA), and via reaction with Clarity™ Western ECL Substrate (Bio-Rad, Hemel Hempstead, UK), following the manufacturer’s instructions. Membranes were read using a Bio-Rad ChemiDoc XR (Bio-Rad, Hemel Hempstead, UK).

### 2.8. Stopped-Flow Light Scattering

Stopped-flow light scattering was performed to measure the membrane permeability of human sperm cells to glycerol, following a method previously described, with slight optimizations for sperm cells [20]. The stopped-flow light scattering experiments were performed using a Stopped-Flow SX20 apparatus (Applied Photophysics, Leatherhead, Surrey, UK), which has a 1 ms dead time, and the temperature controlled at 25 °C. The osmotic shock was performed with a hyperosmotic glycerol solution (isotonic medium supplemented with 250 mM glycerol). A minimum of seven runs were stored and analyzed in each experimental condition. In each run, 100 µL of sperm suspension was mixed with an equal amount of hyperosmotic glycerol solution. The kinetics of the adaptation of sperm volume were measured from the time course of 90° scattered light intensity at 450 nm, until a stable light scatter signal was attained. The isotonic solution osmolarity was determined from freezing point depression on an Osmometer Basic (Löser, Berlin, Germany), using standards of 300 and 900 mOsm. Treatment with AQP inhibitors did not change the spermatozoa volume or viability without an osmotic shock. Since sperm cells are not spherical, the results are presented as the exponential rate coefficient of the volume change, and that can be assumed as proportional to the sperm glycerol permeability.

### 2.9. Statistical Analysis

Experimental results are presented as mean ± standard error of the mean (SEM) if the data followed a Gaussian distribution, or as median [interquartile range] otherwise. Statistical analysis was performed using GraphPad Prism 8 (v8.0.1.244, GraphPad Software, USA). All data sets were screened for outliers using the ROUT method with Q = 1%. The statistical significance of the permeability assay was assessed with the Kruskal–Wallis test, followed by Dunn’s multiple comparison test, as the data do not follow a Gaussian distribution according to the Shapiro–Wilk normality test. If the data passed the normality test, a mixed-effects analysis was performed, with the Geisser–Greenhouse correction, followed by Dunnett’s multiple comparison tests, with individual variances computed for each comparison. Comparisons between normozoospermic and asthenozoospermic were performed using an unpaired *t*-test. *p* < 0.05 was considered significantly different.

## 3. Results

### 3.1. AQP3 and AQP7 Are Localized in Complementary Locations in Human Sperm

The literature available on the localization of the different aquaglyceroporins is still a controversial topic, especially that of AQP7, and particularly in sperm cells. To clarify this information, our first step was to determine the localization of both aquaglyceroporins in human sperm. AQP3 was found in the tail and the base of the head, and was noted to show a distinct lack of signal in the midpiece of the sperm cells (Figure 1A). AQP7 was found to be present in the head and the midpiece of human sperm, with a weak signal in the tail of some spermatozoa (Figure 1B). Thus, both aquaglyceroporins are present throughout the sperm membrane in a complementary fashion. It is worth mentioning that while the AQP3 protein pattern was consistent among all sperm cells analyzed (Figure 1C), the AQP7 protein pattern seemed inconsistent between spermatozoa within a sample from the same individual (Figure 1B,D).

### 3.2. Highly Motile Sperm Cells Have Higher Glycerol Permeability Than Sperm with Low Motility

Gradient separation allowed the separation of the most homogeneous and motile sperm (90% phase) from the immature and lower-motility sperm cells (45% phase) in the normozoospermic samples, and the measurement of their glycerol permeability. Low-motility sperm were less permeable to glycerol (0.84 ± 0.05—fold variation to 90% phase) when compared to high-motility sperm (1.00 ± 0.06—fold variation to 90% phase) (Figure 2A). Despite our results showing a correlation between motility and glycerol permeability in sperm from the same individual, when we opened the research to sperm from asthenozoospermic men versus normozoospermic men, our results did not show such a correlation. The glycerol permeability of the total sperm from the asthenozoospermic (0.94 ± 0.09—fold variation to the normozoospermic group) and normozoospermic samples (1.00 ± 0.09—fold variation to the normozoospermic group) was statistically similar (Figure 2B). Thus, we branched out, and studied the role of each aquaglyceroporin in sperm motility.

### 3.3. AQP7 Is the Primary Aquaporin for Glycerol Diffusion in Human Sperm

Our results show no alterations in sperm motility when exposing sperm cells to the already-mentioned concentrations of specific aquaglyceroporin inhibitors for 10 min in the absence of osmotic stress. Concerning the cells of the control group (1.00 ± 0.21—fold variation to the control group), neither AQP3 inhibition (0.73 ± 0.11—fold variation to the control group) nor AQP7 inhibition (0.89 ± 0.19—fold variation to the control group) had any effect on sperm motility. However, incubation with the general aquaglyceroporin inhibitor phloretin caused a decrease in sperm motility (0.73 ± 0.11—fold variation to the control group), in comparison to that of cells from the control group (Figure 3A), without compromising sperm viability (Appendix A).

Different aquaglyceroporins have different affinities to glycerol, changing their overall permeability to this solute [17]. Thus, we studied the role of each aquaglyceroporin in sperm glycerol permeability after a hyperosmotic shock created by a glycerol solution, with the help of specific AQP3 and AQP7 inhibitors. In sperm from normozoospermic samples, the glycerol permeability was significantly reduced in sperm with inhibited AQP7 (0.39 [0.25;0.59]—fold variation to the control group), in relation to that seen with the AQP3 permeability inhibited (0.64 [0.41;1.13]—fold variation to control group), and the control group (0.82 [0.57;1.37]—fold variation to control group). The incubation of the sperm with the non-selective inhibitor of aquaglyceroporins (phloretin) also reduced the glycerol permeability of the sperm’s membrane (0.42 [0.22;0.59]—fold variation to control group) in relation to the AQP3 inhibition group and the control group, but not to the group where AQP7 was inhibited (Figure 3B). That indicates that AQP7 is the AQP most responsible for glycerol import in sperm from normozoospermic men.

### 3.4. AQP7-Mediated Glycerol Permeability Is Impaired in Sperm from Asthenozoospermic Men

To further explore the link between sperm motility and glycerol permeability, we studied the effect of AQP7 inhibition in sperm from normozoospermic and asthenozoospermic samples. Our results showed that AQP7 inhibition in sperm from asthenozoospermic samples is less effective in decreasing sperm glycerol permeability (1.29 [0.92;2.17]—fold variation to control group) than in sperm from normozoospermic samples (0.87 [0.55;1.31]—fold variation to control group) (Figure 3C). Despite this, our results did not show any difference in the protein expression levels of AQP7 between sperm from normozoospermic samples (1.00 ± 0.19—fold variation to the normozoospermic group) and asthenozoospermic samples (0.87 ± 0.16—fold variation to the normozoospermic group) (Figure 3D). On the other hand, a negative correlation was evident between the effect of AQP7 inhibition on sperm glycerol permeability, and on sperm motility (r = −0.3599) (Figure 3E). That indicates that the spermatozoa with a lower AQP7 permeability (or that have a higher glycerol permeability after AQP7 inhibition, in our results) have a lower sperm motility; therefore, AQP7 could be a key factor in sperm motility.

### 3.5. Sperm Capacitation Is Accompanied by an Increase in AQP7 Levels in Human Spermatozoa

To further study the relation between AQP7 and sperm motility, we studied the levels of this membrane pore during the capacitation process of sperm cells, and the development of hypermotility. The occurrence of the capacitation process was confirmed using tyrosine phosphorylation quantification. The non-capacitated sperm tyrosine phosphorylation (1.00 ± 0.27—fold variation to the non-capacitated group) was decreased, in relation to the one seen in the group of capacitated sperm (2.60 ± 0.32—fold variation to the non-capacitated group) (Figure 4C). A representative blot is demonstrated in Figure 4D.

The AQP7 expression in capacitated sperm (1.53 ± 0.16—fold variation to the non-capacitated group) was higher than in the non-capacitated sperm (1.00 ± 0.08—fold variation to the non-capacitated group) (Figure 4A). Additionally, the increase in AQP7 protein expression resulted in an increase in the total glycerol permeability in capacitated sperm cells (Figure 4C). Non-capacitated sperm (1.00 ± 0.19—fold variation to the non-capacitated group) had a decreased membrane glycerol permeability, compared to capacitated sperm (2.72 ± 0.73—fold variation to the non-capacitated group).

## 4. Discussion

Sperm membrane permeability and, more specifically, glycerol permeability, has been explored in multiple studies. However, most of those studies concern is its impact on animal sperm cryopreservation [21,22]. Thus, the role of sperm glycerol permeability in sperm function has been on the back burner. It is already known that AQP9 is crucial for the diffusion of glycerol in the epididymal epithelia [23]. Interestingly, the lumen glycerol concentration is greater (1.15 mM) than the plasma glycerol concentration (0.35 mM) in rats, suggesting that this glycerol arises from epithelial glycerol-based lipid degradation [13]. It was even found that glycerol-based lipid metabolism disruption in epididymal epithelial cells is linked to male infertility in mice [24]. Taking that into consideration, as well as the fact that multiple studies suggest that sperm cells can metabolize glycerol [14,15,16], we challenged ourselves to enlighten the role of glycerol permeability in human sperm motility.

The first step was to determine the localization of both AQP3 and AQP7 in human spermatozoa. Our results showed that AQP3 is present in the tail and the base of the head of human sperm. AQP3 had already been described in the tail and head of human sperm [8,9]. On the other hand, our results showed a strong signal of AQP7 in the head and midpiece, and a weaker signal throughout the tail of human spermatozoa. Previous studies had already described the localization of AQP7 in human spermatozoa. However, the results were somewhat different between studies. One report showed AQP7 presence inthe midpiece and anterior portion of the tail [10], whereas another study demonstrated AQP7 in the equatorial segment of the head, pericentriolar area and midpiece, with weak labeling in the tail [11]. Yet another showed AQP7 localization only in the head of human sperm [9]. As referred to previously, while AQP3 expression was consistent within all the sperm cells analyzed, the AQP7 expression pattern seemed inconsistent between spermatozoa within a sample from the same individual, which might explain the conflicting results described in previously published studies. This heterogeneity can arise due to differences in the developmental or maturation process of sperm cells during their transit through the epididymis, where they acquire motility and functional changes. This maturation process can involve modifications in the protein amount, and post-translational modifications. The timing and extent of these changes may vary among individual sperm cells, leading to inconsistent protein expression patterns [10].

The compartmentalized nature of aquaglyceroporin expression in human sperm is an interesting concept and worth exploring. It opens up the possibility that different aquaglyceroporins might have different roles in glycerol permeability. To assess that, we used specific AQP3 and AQP7 inhibitors to unravel the relevance of each aquaglyceroporin in the glycerol permeability of sperm from normozoospermic men [17]. Our results showed that AQP3 inhibition did not have any effect on the sperm glycerol permeability. However, AQP7 inhibition decreased the sperm glycerol permeability by 55%; a similar magnitude is seen in the group treated with phloretin, the general aquaglyceroporin inhibitor. Thus, AQP7 appears to be the main aquaglyceroporin responsible for the facilitated diffusion of glycerol into sperm cells from normozoospermic men. This tendency was also noted in different cell types, in which AQP7-expressing cells were shown to have a considerably higher glycerol permeability than AQP3-expressing cells [17]. On the other hand, AQP3 has been suggested to be an important osmosensor, specialized in the efflux of water out of the cell after a hypotonic osmotic shock, in the tail of mice sperm [8]. These results could indicate that while AQP7 is more devoted to mediating the import of glycerol, AQP3 might be more permeable to water, meaning that both aquaglyceroporins demonstrate a complementary function and localization in human sperm. It should also be considered that AQP3 is also a peroxiporin, unlike AQP7. Peroxiporins can play a role in hydrogen peroxide diffusion, which triggers various signaling pathways in sperm capacitation [7,25,26]. Nevertheless, it is worth mentioning that AQP7 has a strong presence in the head and midpiece of human sperm, and that the change in sperm volume noted by the stopped-flow light scattering apparatus could be higher in magnitude than that seen in the tail of human sperm, where AQP3 is mainly expressed.

Taking into consideration the results described, an interesting comparison can be made. AQP9 is the main aquaglyceroporin in the liver, as it is in the male reproductive tract epithelia. However, AQP9 is known to be mainly responsible for the import of glycerol into the liver, while being the main actor in cellular glycerol export in the male reproductive tract epithelia into the lumen [12,27,28]. On the other hand, AQP7 is thought to be the main actor in glycerol efflux in adipose tissue [29,30,31] whereas, according to our results, it is the main actor in glycerol influx in human sperm. This is one more indication that aquaglyceroporins are versatile in their role. It is also fair to assume that, as with aquaglyceroporins in the adipose tissue and liver, sperm aquaglyceroporins could also have a role in lipid/fatty acid metabolism. The literature on this subject is still scarce, but glycerol has been found to be used in de novo phospholipid synthesis in bull sperm [32]. The role of lipids and glycerol on sperm bioenergetics is still a topic of discussion. Sperm cells have been found to express the machinery necessary for the fatty acid β-oxidation [33]. Further studies also point to a significant role of the fatty acid β-oxidation on ATP production and sperm motility [34,35,36]. However, the linkage between sperm aquaglyceroporin expression and lipid or fatty acid metabolism still eludes the scientific community, but is certainly something worth exploring.

Many have agreed that AQP7 expression is a good indicator of sperm quality [10,11]. Herein, we tried to study the relationship between sperm motility and glycerol permeability within the same individual, by separating high-motility sperm, and low-motility and immature sperm, through gradient centrifugation. Our results showed that the high-motile phase spermatozoa presented a higher glycerol permeability than the low-motile spermatozoa phase. Previous reports showed that AQP7 expression is also downregulated in sperm from the low-motile phase after a similar protocol of sperm separation [11]. One must bear in mind that the low-motility phase also has a higher level of immature spermatozoa, compared to the high-motility sperm phase, which could influence (to some extent) the results obtained in the stopped-flow light scattering device. Contrastingly, with these results having been obtained from the same individual, we were able to remove all lifestyle and comorbidity variants that are impossible to escape when comparing different individuals. Regardless of this, we demonstrated that AQP7-mediated glycerol permeability is decreased in sperm from asthenozoospermic men, compared to sperm from normozoospermic men, despite the same levels of AQP7 being presented between cells from both groups. The different results seen between function and expression are interesting, and should be further explored in future works. One possible reason for this mystery may lie in the aquaglyceroporin interactome [37]. It is interesting to note that the total sperm membrane glycerol permeability is equal between the sperm from normozoospermic and asthenozoospermic men, but this is not the case for the AQP7-mediated glycerol permeability, which is reduced in asthenozoospermic men. In fact, our results indirectly indicate that AQP7 glycerol permeability is positively correlated to sperm motility.

As stated above, AQP7 expression in human sperm is a target of some scientific discussion, as the literature shows that its pattern can change among and within individuals. As the expression pattern of AQP7 changes during spermatogenesis, more specifically during spermiogenesis [10], it could be fair to assume that some dysregulation in this process can account for these changes. On the other hand, the AQP7 pattern changes from testicular to ejaculated sperm [10], possibly during sperm maturation, as referred. Fish homolog AQPs have also been shown to be able to be translocated from the sperm membrane of seabream to new membrane sections after capacitation [38], emphasizing the dynamic nature of AQP expression. Sperm protein synthesis is another topic that has been discussed for decades, but each year it becomes harder not to acknowledge it. Multiple studies have indicated that sperm can perform de novo protein synthesis, especially during capacitation [39,40,41,42]. Our results showed that AQP7 might be one of the proteins that are upregulated during that process. This increase in AQP7 expression is accompanied by the development of sperm hypermotility, comprising one more link in the relationship between AQP7 and sperm motility. The increase in AQP7 expression did translate into an increased sperm membrane glycerol permeability, in our results. The sperm plasma membrane undergoes structural changes during capacitation that theoretically render it less stable and more fluid [43]. Evidence suggests that this process aids the development of hypermotility, and facilitates the frantic tail movement that is characteristic of this capacitation phase [44,45]. Perhaps glycerol could have an important role in the final stages of the sperm’s journey. The removal of human sperm from a glycerol-containing medium, after cryopreservation, to a glycerol-free medium was described to decrease the motility and capacitation rate [46]. On the other hand, female arousal is accompanied by an increased vaginal glycerol secretion [47], which is thought to be responsible for increasing lubrification. As multiple aquaglyceroporins have already been confirmed throughout the female reproductive tract [12], it would be pertinent to study whether the secretion of glycerol also increases upstream of the vaginal canal, where lubrification is less important, which could indicate another symbiotic process between the female reproductive tract and the sperm cells, to potentiate fertilization. However, this is just a theory, and clear studies are necessary to fuel this hypothesis. It is also worth noting that this study was performed with samples that were all obtained from a fertility clinic, and that our sample does not represent the global male population, due to the criteria for sperm sample selection that we applied. Moreover, the measurement of glycerol permeability is only possible assuming that the entry of water and glycerol in the sperm cell is proportional to the change in cell volume, and that the latter is proportional to the intensity of the light scattering. The sperm cells are highly specialized, and thus do not have the best shape for this type of measurement, not allowing the calculation of glycerol permeability *per se*. Herein, we compared the exponential rate coefficient between the different groups, assuming that the viability of the sperm is similar between samples, as all of them show a normal and similar sperm viability. One limitation of this method is that we did not measure or classified the morphology of the sperm cells within our samples. It has already been reported that sperm from asthenozoospermic samples have a higher probability of having tail defects than those from normozoospermic samples [48,49]. It is possible that such morphological changes can influence the results of the glycerol permeability measurement.

In conclusion, our results indicate that AQP7 is the aquaporin responsible for glycerol permeability in sperm from normozoospermic men, and that this mechanism is impaired in sperm from asthenozoospermic men. However, that could lead us down two different routes: glycerol diffusion is important for sperm bioenergetic and thus, sperm motility; or, AQP7 is the key modulator of unknown pathways or mechanisms. With each passing year, there is more information regarding AQP7 and aquaglyceroporins, not only as pores for small solutes but as real contributors to signaling pathways. That could be the reason for the results herein obtained; however, more work is needed to study such events.

## Figures and Tables

**Figure 1 cells-12-02003-f001:**
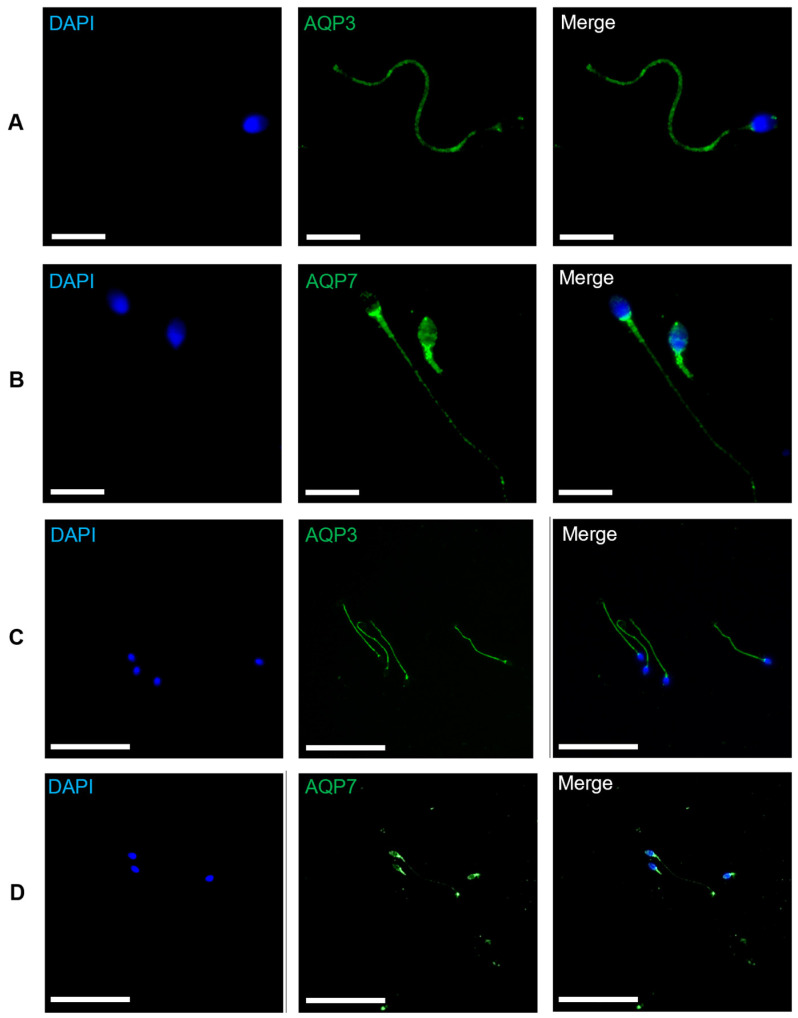
Immunofluorescence staining of AQP3 and AQP7 in human sperm. Green immunolabeling of AQP3 (**A**,**C**) and AQP7 (**B**,**D**) at two different amplifications. Negative controls were carried out, without a primary antibody, for each protein. Cell nuclei were stained blue with VECTASHIELD^®^ Antifade Mounting Medium with DAPI. (**A**,**B**) White bar, 10 μm; (**C**,**D**) white bar, 50 μm.

**Figure 2 cells-12-02003-f002:**
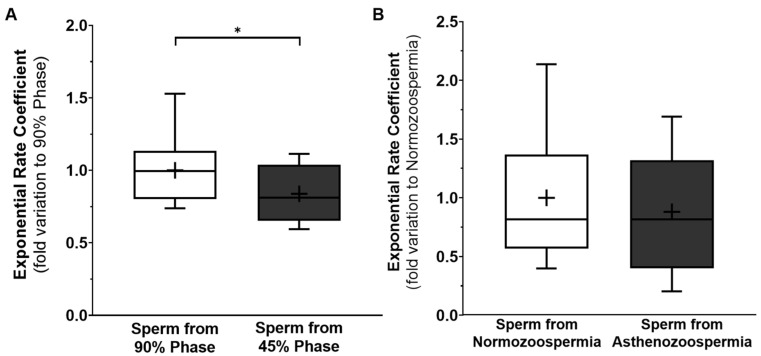
Membrane glycerol permeability (exponential rate coefficient) of human sperm from 90% phase versus 45% phase, after gradient sperm separation (**A**); and sperm from men with normozoospermia versus sperm from men with asthenozoospermia (**B**). Results presented in min. to max. graphs, with a cross representing the mean of the sample size. Significantly different results (*p* < 0.05) are indicated as: *. (**A**) *n* = 12; (**B**) *n* = 30.

**Figure 3 cells-12-02003-f003:**
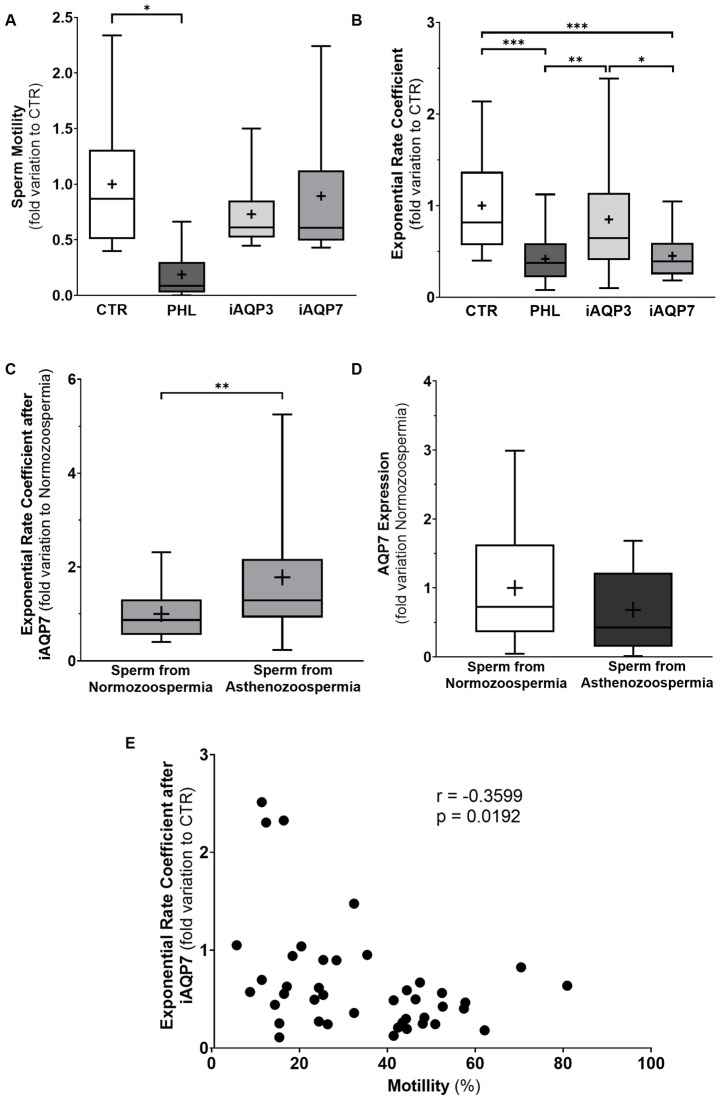
Representation of the effect of AQP3 and AQP7 inhibition with 200 nM of Z433927330 (iAQP3) and DFP00173 (iAQP7), respectively, and both AQGPs with phloretin (PHL) in sperm motility (**A**) and membrane glycerol permeability (exponential rate coefficient) (**B**) of spermatozoa from normozoospermic men. Effect of AQP7 inhibition in membrane glycerol permeability (**C**) and AQP7 expression levels (**D**) of sperm from normozoospermic versus asthenozoospermic men. Results presented in min. to max. graphs, with a cross representing the mean of the sample size. Significantly different results with *p* < 0.05 are indicated as: *; *p* < 0.01 are indicated as: **; and *p* < 0.001 are indicated as: ***. Graphical representation of the negative correlation between the effect of AQP7 inhibition on sperm glycerol permeability and the percentage of sperm motility (**E**). (**A**) *n* = 12; (**B**) *n* = 30; (**C**) *n* = 30; (**D**) *n* = 20; (**E**) *n*= 42.

**Figure 4 cells-12-02003-f004:**
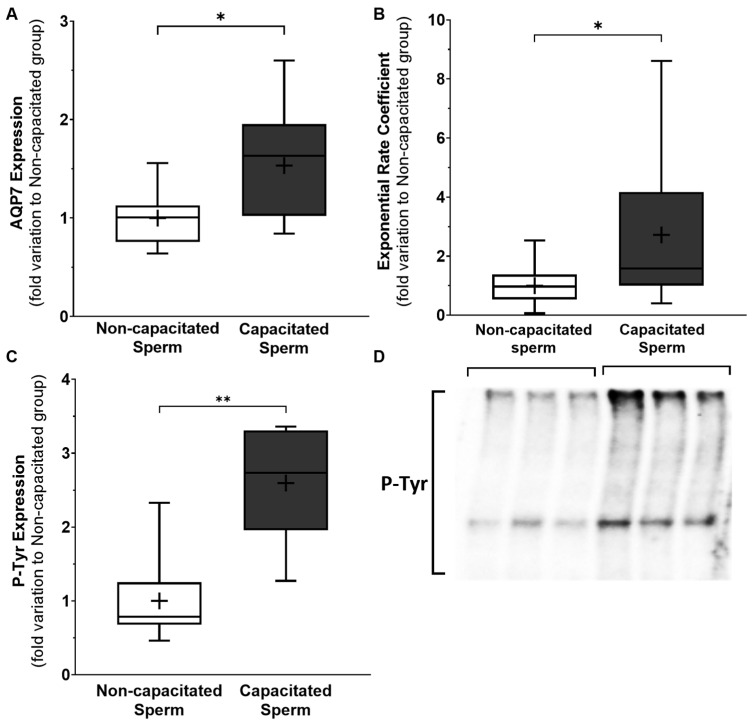
Protein expression levels of AQP7 (**A**), membrane glycerol permeability (exponential rate coefficient) (**B**), level of tyrosine phosphorylation (p-Tyr) (**C**), and representative blot (**D**) of non-capacitated sperm versus capacitated sperm. Results presented in min. to max. graphs, with a cross representing the mean of the sample size. Significantly different results with *p* < 0.05 are indicated as: *; *p* < 0.01 are indicated as: **. *n* = 12.

**Table 1 cells-12-02003-t001:** Characterization of samples from normozoospermic and asthenozoospermic men.

	Concentration (10^6^ Cells/mL)	Viability (%)	Motility (%)
Normozoospermia	117.8 ± 15.94	70.20 ± 1.54	49.89 ± 1.63 *
Asthenozoospermia	112.8 ± 17.60	66.40 ± 1.21	19.93 ± 1.34 *

* *p* < 0.05.

## Data Availability

The data presented in this study are available on request from the corresponding author. The data are not publicly available, due to privacy and ethical restrictions.

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
