# Peer review of "Aquaporin-7-Mediated Glycerol Permeability Is Linked to Human Sperm Motility in Asthenozoospermia and during Sperm Capacitation"

_cells, 2023, doi:10.3390/cells12152003_

Round 1

Reviewer 1 Report

The MS entitled ‘submitted by Ribeiro et al. reports the investigation of AQP-mediated glycerol permeability in sperm motility and hypermotility during sperm capacitation. The authors have collected semen samples from 30 normozoospermic individuals (sperm concentration ≥ 15 million/mL and sperm 87 viability ≥ 58%) and from 30 asthenozoospermic men (total motility was < 40%). They have analyzed AQP3 and AQP7 protein amount and activity in high- versus -low motile fractions of spermatozoa and in capacitated and non-capacitated spermatozoa. In particular by using specific and non-specific inhibitors of AQP, the authors show that AQP7 but not AQP3 is essential for glycerol permeability in human spermatozoa. The authors also show that that selected high-motile spermatozoa have a higher glycerol permeability than the low-motile spermatozoa. They also show that sperm cells from asthenozoospermic men have less AQP7 activity than those from normozoospermic men while the protein amounts were found equal. Finally, they show that AQP7 protein amount was increased in capacitated sperm compared to non-capacitated sperm

The study reported by the authors is novel and interesting. It provides information regarding glycerol permeability in human sperm cells, which is of interest in the context of potential metabolization of glycerol for energy production. This point shouldbe more stressed by the authors in the discussion. In general, the MS would need careful reading and English language editing (see below).

Major comments

1. The authors should provide the sperm motility parameters of the normozoospermic population in the M&M section. Could they clarify in the selecting criteria that the total sperm motility for the normozoospermic men was > 40% (Line).

2. The authors should clarify why only 20 samples among the 30 initially selected for each category (normozoospermic and asthenozoospermic) were further analyzed.

3. The authors should provide the percentage of identity/homology between AQP3 and AQP7 and also provide the epitopes, which were used to raised the antibody. The absence or possible existence of cross reaction for both antibodies should also be discussed.

4. Line 250: the authors claim that ‘Our results show no alterations in sperm viability or motility when exposing sperm cells to the specific aquaglyceroporin inhibitors for 10 minutes in the absence of osmotic stress.’ The authors should provide methodological information for these results: specification of the inhibitors which were used, their concentrations and the number of spermatozoa. The authors should also illustrate these negative results as supplemental data.

5. Considering the localization of AQP3 and AQP7 in sperm head, which is evidenced by the authors, the analysis/correlation of acrosomal reaction with glycerol permeability and AQP protein amounts would be interesting. Did the authors perform acrosomal staining and evaluated the AR rate in presence of AQP inhibitors (staining with fluorescent PNA or PSA for instance)? Is there a correlation between AQP3 and AQP7 protein levels and activities with AR rate (spontaneous AR and induced AR)?

6. As glycerol is a metabolite linking glycolysis and fatty acid metabolism, the importance of glycerol permeability in spermatozoa and its increase after capacitation could be discussed by the authors, .

English Language editing

In all the manuscript, please change ‘expression’ by ‘protein amount’ when discussing AQP7 protein distribution in sperm cells

Idem ‘P-Tyr expression’ should be changed by ‘P-Tyr level’.

1. Line 43:

Please correct ‘This phenomenon allows the sperm cells to move through the female reproductive tract and there, after capacitation, develop the state of hypermotility for the final stages of its journey by ‘This allows the sperm cells to move through the female reproductive tract and to develop the state of hypermotility at the final stage of their journey, an activation process called capacitation.

The above sentence should also be placed after the sentence ‘Sperm cells acquire their motility during the maturation process in the epididymis, where the medium osmolarity is in constant change throughout its different sections [3].’

2. Line 47:

Please correct ‘Thus, sperm osmoregulation is 47 important for proper sperm maturation [4-6].’ By ‘As a result, osmoregulation is important for proper sperm maturation [4-6].’

3. Line 51:

Please removed ‘have been identified’

4. Line 56:

Please correct the sentence as follow ‘The role of glycerol’

5. Line 66:

Please correct the sentence as follow ‘Considering that sperm motility is acquired during epididymal transit’

6. Line 76:

Please correct the sentence as follow ‘Semen samples were obtained from male patients by masturbation after 2-4 days of abstinence and placed in sterile tubes’

7. Line 79: The sentence ‘All patients included in this study signed informed written consent.’ Should be placed at the beginning of the paragraph (line 73).

8. Line 83:

Please remove ‘were’

9. Line 86:

Please remove ‘only’ and also specify that these samples are the samples which were categorized as normozoospermic

10. Line 286: Please complete the following sentence with a verb ‘On the other hand, a negative correlation between the effect of AQP7 286 inhibition on sperm glycerol permeability and sperm motility (r = -0.3599) (Figure 3E).’

In general, the MS would need careful reading and English language editing (see requested modifications).

Reviewer 2 Report

The paper reports an interesting study on the expression and role of AQP3 and AQP7 in human spermatozoa. The results indicate that AQP7 is the aquaporin responsible for glycerol permeability in spermatozoa of normozoospermic sperm and that this mechanism is impaired in sperm from asthenozoospermic men. In addition, AQP7 expression increased in capacitated sperm compared to non-capacitated sperm. Authors concluded that glycerol diffusion could be an important process for sperm bioenergetic and motility.

The paper is well written and the methods are appropriate.

1)     The most important flaw of the study regards the evaluation of sperm morphology that was completely omitted. The authors dismiss this problem with a sentence in the discussion stating that “sperm morphology is similar among samples as they all have a normal and similar sperm morphology". The high degree of sperm morphological variability is well known; for instance, the asthenozoospermic patients included in this study most likely showed sperm with coiled tails (coiled tails are almost always present and more frequently visible in asthenozoospermic men) and the morphology cannot be compared to that of normozoospermic men.

I believe that sperm morphology should be reported in table 1 as well, and the authors should discuss the issue of morphology while citing other studies.

2)     Lines 81-83: Was the analysis of the semen performed after PBS washing? Semen analysis must be performed following the liquefaction of the raw semen. The authors should clarify this point.

3)     Lines 107-108: “A total of 100 sperm cells were counted per sample/condition and the number of sperm with tail movement was regarded as motile sperm”: please the authors should clarify which type of motility they considered: rapid progressive, slow progressive, non-progressive…

4)     Table 1: authors should add the sperm morphology parameter. In addition, how many patients have been seen to find samples with such high concentration and low motility? It is quite common that concentration and motility (and morphology) are both reduced.

5)     Statistical analysis: if the data are not normally distributed, the data should be reported as median and intequartile range (in the boxplot the line in the box is generally the median)

6)     There are typing errors in the graphs: “exponencial” is exponential (figs 2, 3, 4…), motility (fig 3); please the authors should check carefully all the graphs

7)     Line 315: in the text “fig 4C” is related to P-tyr expression and not to glycerol permeability. Please the authors should check carefully the correspondence between the text and the quoted figures.

In discussion the major problem concerning sperm morphology should be addressed.

Round 2

Reviewer 1 Report

The authors have answered all my points and provided the requested complementary information.

Reviewer 2 Report

The paper is improved and can be accepted